# Mapping Spreadsheets to RDF:
# Supporting Excel in RML

Markus Schröder, Christian Jilek, and Andreas Dengel

[1] Smart Data & Knowledge Services Dept., DFKI GmbH, Kaiserslautern, Germany
[2] Computer Science Dept., TU Kaiserslautern, Germany
{markus.schroeder, christian.jilek, andreas.dengel}@dfki.de

**Abstract.** The RDF Mapping Language (RML) enables, among other formats, the mapping of tabular data as Comma-Separated Values (CSV) files to RDF graphs. Unfortunately, the widely used spreadsheet format is currently neglected by its specification and well-known implementations. Therefore, we extended one of the tools which is RML Mapper to support Microsoft Excel spreadsheet files and demonstrate its capabilities in an interactive online demo. Our approach allows to access various meta data of spreadsheet cells in typical RML maps. Some experimental features for more specific use cases are also provided. The implementation code is publicly available in a GitHub fork.

**Keywords:** Spreadsheet · Excel · RML · RDF · Knowledge Graph

## 1 Introduction

As soon as knowledge graphs have to be constructed from given data, the use of mapping languages is a common practice. Such languages let users define declarative rules to map various input data formats to a complex interconnected graph. In the Semantic Web community such graphs are usually modeled with the Resource Description Framework (RDF) [11] and (semi-)structured data is mapped with the RDF Mapping Language (RML) [2] – a superset of the W3C-recommended mapping language R2RML [10]. The specification of RML [1] describes how semi-structured data, like XML or JSON, and tables in form of Comma-Separated Values (CSV) can be mapped properly. However, tabular data can also be found in other file types, prominently in spreadsheets.

Since the spreadsheet methodology enables a well understood, easy and fast possibility to enter data, they are widely used by knowledge workers, especially in the industrial sector. In contrast to simply structured CSV files, spreadsheets can model complex workbooks containing multiple sheets with meta data rich cells. Besides it content, a single cell may store additional information like its appearance (colors, styles and borders) or cell comments. How these cells should be filled or styled by users is not predetermined by spreadsheet applications in general. As a consequence, in practice, cells can contain inconsistent and unstructured content which can be arbitrarily arranged in a sheet. That is why a mapping of such data to RDF can become a challenging task.

Although, spreadsheets are frequently used in industry, well-known RML-supporting tools like RML Mapper[3], CARML[4], RocketRML [9], SDM-RDFizer [4] or Mapeathor [5] currently do not support them natively. To fix this issue, we extended RML Mapper to support Microsoft Excel spreadsheet files. We have chosen RML Mapper because its code structure let us easy embed our new code. Natively supporting spreadsheets in RML has several advantages. There is no need anymore to preprocess and transform spreadsheets in a format that can be handled by a mapping tool (e.g. CSV). This removes the otherwise additional effort for mapping experts and let them focus on defining proper rules. Having RML rules that directly refer to spreadsheet data makes interpretation of rule definitions straightforward. Thus, it becomes easier to communicate to a data provider how their spreadsheets will be mapped to a knowledge graph. Additionally, practitioners of RML who already learned the language become able to map spreadsheets with almost no extra effort.

In the next section we describe in detail how we realized the Excel support in RML.

## 2   Supporting Excel in RML

Our approach is implemented as an extension to the RML Mapper tool in a separate GitHub fork[5]. For demonstration purpose, we additionally provide an interactive demo page[6] where visitors can try various mapping examples.

At its core, our component utilizes the Apache POI[7] library to read Microsoft Excel spreadsheets. In order to refer to spreadsheet contents as a logical source, a small spreadsheet ontology[8] (usually prefixed with `ss`) was designed. As demonstrated in Listing 1.1, using a spreadsheet reference formulation (Line 2), a workbook source should name a spreadsheet file (Line 5), a referring sheet by name (Line 6) and a range of cells in the sheet (Line 7). A triples map that uses this logical source will iterate over single cells in the given range.

Listing 1.1: Exemplary RML definition of a spreadsheet file as a logical source.

```
1  [ a rml:LogicalSource ;
2    rml:referenceFormulation ql:Spreadsheet ;
3    rml:source [
4      a ss:Workbook;
5      ss:url "workbook.xlsx" ;
6      ss:sheetName "Papers" ;
7      ss:range "A2:A5" ;
8      ss:javaScriptFilter "/Know\\w*/.test(valueString)" # optional
9    ]
10 ]
```

---

[3] `https://github.com/RMLio/rmlmapper-java`

[4] `https://github.com/carml/carml`

[5] `https://github.com/mschroeder-github/rmlmapper-java/tree/mschroeder-features`

[6] `http://www.dfki.uni-kl.de/~mschroeder/demo/excel-rml`

[7] `https://poi.apache.org/`

[8] `http://www.dfki.uni-kl.de/~mschroeder/ld/ss`

Optionally, a filter can be added that picks certain cells based on a JavaScript program (Line 8). Regarding our example, a regular expression is used to iterate over only those cells which contain the phrase "Know". Using the expressive script language and variables that represent cell meta data, appropriate filter procedures can be realized.

Listing 1.2: Demonstration of how cell meta data can be accessed in RML maps.

```
1   rr:subjectMap [
2     rr:template "http://example.org/{address}"
3   ] ;
4   rr:predicateObjectMap [
5     rr:predicateMap [
6       rr:template "http://example.org/{[2,0].valueString}"
7     ] ;
8     rr:objectMap [
9       rml:reference "(2,0).valueNumeric"
10    ]
11  ]
```

In various RML maps, meta data of the currently iterated cell can be retrieved, as demonstrated in Listing 1.2. For example in Line 2, a template expression inserts the cell's address (like "A2") to form a unique URI for a subject resource. However, often it is required to refer to nearby cells in order to construct appropriate statements. To refer to other cells relatively from the current cell, we introduce a parenthesis notation, like `(column,row)`. A column shift (x-axis of the sheet) and a row shift (y-axis of the sheet) allows to reach any cell relative to the cell representing the subject resource. Regarding Line 9 of our example, numeric values which are located two columns away on the right are used as objects in the mapped statements. In a similar way, cells can be referenced absolutely by using square brackets (Line 6).

As already mentioned, spreadsheet cells have several meta data values that need to be retrievable in a mapping through variables (written in the following with monospaced font). A cell's location in a sheet can be accessed either as a usual spreadsheet `address` or as `column` and `row` indices. Since a cell (if not empty) stores either a string value or a numeric value, various possibilities are given to access its content: `valueNumeric` retrieves a floating point value (`valueInt` an integer value), `valueBoolean` a boolean value, `valueFormula` if a cell contains a formula (`valueError` to get its possible error code) and `valueString` retrieves its text content. Besides content, one could also be interested in the appearance of a cell in a sheet: since it can be colored, `backgroundColor` and `foregroundColor` can be queried as a hexadecimal RGB value. Regarding the used font, `fontColor`, `fontName` and `fontSize` are available. On a more fine-grained level, cells can also store formatted text which is returned by `valueRichText`. The formatted text is represented in an HTML-like syntax, for example,
"<font face='Arial' color='#ff0000'>red, italic and bold</font>".

If spreadsheets were completed in an inconsistent manner, it happens that cells were unintentionally filled with different data types. For such cases, one can use the `value` variable to always obtain a string representation of a cell's content regardless of its cell type. However, if this is not sufficient because more details are needed, the `json` variable could be used to retrieve a JavaScript Object

Notation (JSON) representation of a cell. The JSON object contains, besides the cell type, various data types mentioned above.

### 2.1    Experimental Features

We also would like to propose some experimental features that we found useful in our use cases. However, the official introduction of these extensions would require to change the RML specification on some points.

**Multiple Different Properties in a Cell.** In our use cases, we frequently experienced that users record multiple information in one cell. Each piece of information $i_j \in I$ then corresponded to a different property $p_j \in P$. Following the current RML specification, one has to define for each $p_j \in P$ a separate predicate-object map. Instead, we propose a shortcut such that only one predicate-object map is needed. This is done by allowing that an RDF list of properties $p_j \in P$ can be passed in a predicate map. By implementing a suitable Function Ontology ($F_nO$) [7] procedure, we extract information pieces $i_j \in I$ from a cell's content and return them as an object list $o_j \in O$. Usually, RML provides that a Cartesian product is formed between predicates and objects which is $P \times O$. However, in our case we need to zip predicate and object lists such that the following set is made: $\{(p_j, o_j)\}$. Thus, instead of all possible pairs of the $P \times O$ matrix, only the diagonal ones are selected. This new behavior is activated by adding a `ss:zip true` statement to a predicate-object map.

**Multiple Complex Entities in a Cell.** Similarly to the previous observation, we often discovered that users mention several entities in a single cell. A prominent example is a list of persons having first and last names, for instance book authors. Represented as an RDF graph, such complex entities potentially require several statements to be fully expressed. An $F_nO$ function that is able to perform entity extraction would need to return an arbitrary large RDF graph in a certain serialization format (e.g. Turtle) instead of a single value. In order to integrate such a return value in an object map, we define a new term type `ss:Graph`. Once this term type is chosen, the returning RDF graph is parsed and added to the emerging knowledge graph. By using a special `ss:SelectedObjects` resource together with `rr:object` statements, only selected objects will be mapped in an object map.

## 3    Related Work

In the past, several similar approaches were implemented that map spreadsheets to RDF using a language. Domain specific languages (DSL) other than RML are provided to let knowledge engineers express how input data shall be mapped to a graph structure.

Spread2RDF[9] uses a Ruby-internal language while $M^2$ (Mapping Master) [8] builds upon a compact syntax for OWL ontologies (Manchester syntax).

---

[9] https://github.com/marcelotto/spread2rdf

Sheet2RDF [3] uses a special ProjEction of Annotations Rule Language (PEARL), whereas XLWrap [6] utilizes template graphs together with a special expression language to refer to contents of sheets. A different approach is followed by TabLinker[10] which requires that input spreadsheets are annotated in advance with certain styles.

With our implementation, we combine the feature of mapping spreadsheets with the advantages of the well specified RML approach. Those who already used the language and need to process spreadsheets do not have to look for an alternative anymore.

## 4    Conclusion and Outlook

In this paper we discussed that spreadsheets, despite being widely used, are still not supported by the RDF Mapping Language (RML). We therefore proposed a first solution which extends the already existing RML Mapper tool with necessary components. Additionally, experimental features were introduced too. A publicly available web page[11] demonstrates the new capabilities.

In the future, we plan to integrate our proposed solution in the RML specification [1] so that other RML tools may support spreadsheets too. Further, we intend to run several performance and compliance tests with our current reference implementation. This could be done in the R2RML implementation report[12].

***Acknowledgements*** This work was funded by the BMBF project SensAI (grant no. 01IW20007).

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
