# OpenReview forum: "Mapping Spreadsheets to RDF: Supporting Excel in RML"
_eswc-conferences.org/ESWC/2021/Workshop/KGCW — KGCW 2021_

### Official Review · ~Oscar_Corcho1 · 2021-03-24
**Interesting approach to deal with Excel spreadsheets**

**Rating:** 6
**Confidence:** 2

**Review:**

This is an interesting approach for dealing with spreadsheets written in Excel rather than CSVs. The authors have created the corresponding code to deal with this, and added some specific functions that are not common in RML and its related tools to deal with some of the intricacies of Excel spreadsheets.

For a workshop like this one, this is a nice contribution that will surely spark discussions on whether this is a valid approach, whether users should be or not be forced to use CSV or alike instead of Excel, whether all the typical Excel features and ways in which data is normally represented are considered, etc. In any case, as it is a workshop contribution that shows the feasibility of the approach and provides a demo, IMO this is an interesting contribution to accept.

What if we wanted to go further into this area and provide more scientific contributions? I would recommend the following:
- Making an analysis of how data is commonly presented in Excel files across organisations, so as to gather a clear overview of how data is usually represented and which functionalities should be provided in this tool/approach if we want to get a larger coverage.
- Considering how this can be converted into patterns. A work from one of the R2RML contributors from our own research group comes to my mind inmediately in this respect (the PhD thesis of Boris Villazón-Terrazas). I think that this is a must-read if you want to move forward and continue through this line.
- A clear analysis of whether the claim that this avoids knowledge workers going through an additional step of CSV creation is actually right or not. I have the feeling that there is something more than this. CSVs are a bit more structured than Excel, in the sense that whenever somebody is exposing in CSV, a more tabular structure is used, while Excel files may also go into how information is presented (as discussed by the authors) and then more needs may arise. Does it make sense to have special tools that do the transformaiton and then this work may not be so necessary? An interesting question to address.

So, in summary, an interesting contribution for a workshop.

Only a minor comment.. Mapeathor (a tool where I have been involved) is considered as a KG creation tool. I would rather refer to it differently, since this is a tool that uses spreadsheets for the configuration of mappings, so as to ease (in principle, yet to be tested more deeply) the process of RML mapping geneation. This should probably be refined in the paper if accepted.

---

### Official Review · ~Ben_De_Meester1 · 2021-04-13
**Worthwile contributions that are superficially described**

**Rating:** 7
**Confidence:** 5

**Review:**

This paper -- somewhere between a system/demo paper and a short research paper -- describes extensions on top of RML to support Excel spreadsheets, and introduces some more extensions along the way to cover more encountered edge cases.
Many of the proposed changes feel relevant and useful,
however, the paper does not give many insights except for introducing those extensions, gives little argumentation into why certain choices were made,
nor helps the reader understand which technical hurdles were taken.
I'm quite confident the author is very knowledgeable and can well explain the choices made, but sadly, this isn't very clear from the current text.
That said, I think this paper would be a great addition to the workshop
since the work seems very sound and worthwhile,
and I'm looking forward to the discussions 🙂.
If possible, I would like the author to clarify which type of paper he submitted and maybe make (small) changes accordingly, to manage expectations of the reader.

### Detailed remarks

- general remark: it was not clear to me whether this was a system/demo paper or a short research paper. It felt a bit in-between. For a system/demo paper, I would have expected some more technical details (what are the components of the demo, how was the development process, any lessons learned there), for a research paper, I would have expected some more argumentation as to why certain choices were made, among which alternatives.
- The paper misses a clear listing of all contributions, it is now hard to summarize all the different extensions you did on top of RML.
  - Spreadsheet formulation
    - I miss a discussion on how this can or should or should not be extended to other spreadsheet formats (LibreOffice, Google Spreadsheets)?
  - Javascript filter: why like this, since you're already using FnO?
  - valueRichText: why an _HTML-like_ format? Or do you actually mean <https://en.wikipedia.org/wiki/Rich_Text_Format> ?
  - By using a special `ss:SelectedObjects` resource together with `rr:object` statements, only selected objects will be mapped in an object map. --> I had to check the source code what this meant, so that's not clearly described in the paper imo. Also: why put this burden on the function? That means that it becomes very much tied to RML, no? Why did you choose this option?
- Related work: I wonder what's the relation between your related work and your proposal: eg where the fields of the query formulation based on related work or?
- A spec would be nice (but I'm not the one to talk 😅)
- Personally I'm very interested how the devevelopment experience on the RMLMapper went: how did it go, what would you change, what can we learn from your voyage?
- I absolutely love the demo, clear examples with descriptions, interactive. Everyone should do it like this, thanks.

---

### Official Review · ~Vladimir_Alexiev1 · 2021-04-14
**Solid piece of Java engineering work, but the chosen semantic representation approaches are not optimal**

**Rating:** 7
**Confidence:** 5

**Review:**

## Positive
- The paper extends RML Mapper with access to Excel data. It represents a solid piece of Java engineering work, although the chosen semantic representation approaches are not optimal.
- The formatted text of a cell is available with valueRichText. However, it is known that it's hard to control the quality of HTML resulting from Microsoft Office, eg the tag <font face='Arial'> is output in a repetitive way. Font size, color etc are available. Other cell features (eg borders) are not available.
- A demo page http://www.dfki.uni-kl.de/~mschroeder/demo/excel-rml/ is provided, including 12 examples, which is great.

## Major Concerns
- Describe and comment on scalability aspects: what is the biggest number of rows and columns that Excel can handle, whether your tool supports streaming (obviously not), what is the memory consumption for eg an Excel of 500 columns and 1M rows. I see this mentioned in future work ("outlook") but I think it's necessary to comment what are the scale limitations of your tool so users can assess quickly whether it's suitable for a particular real-world case
- Relative references from the current cell are used, eg rml:reference "(3,0).valueBoolean" or rml:reference "(0,0).valueRichText". It appears the tool cannot use Excel headers, and instead uses cell address ranges and relative references. This is a major shortcoming since it makes the conversion scripts:
  - Less declarative: a number is not as telling as a column name; eg referring to a header as "{[1,0].valueString}" (example 7) is unreadable compared to eg "{paper}"
  - Brittle in the face of rearranging the sheet format; eg inserting a column in the middle will break all references on the right of it
  - I would not use the tool because of this shortcoming alone
- "This removes the otherwise additional effort for mapping experts and let them focus on defining proper rules. Having RML rules that directly refer to spreadsheet data makes interpretation of rule definitions straightforward": This is hardly a fair statement, given the above criticism, and eg looking at examples with fnml:FunctionMap
- If the user must provide "ss:url", doesn't that mean that they canot specify the input on the command line?

## Minor Concerns
- Summarize what is the benefit of your tool compared to a unix pipeline using a ready tool like "csvtk"
- If possible, please obtain a host name for the demo server http://173.212.240.179:6982/#
- Serve the ontology in several formats (Turtle, RDF/XML, JSON-LD) with content negotiation. Currently it's served only in Turtle and with wrong content type:
       curl -I -Haccept:text/turtle http://www.dfki.uni-kl.de/~mschroeder/ld/ss
       Content-Type: text/plain

- Fix Unicode problems in the ontology (dc:creator "Markus SchrÃ¶der")
- Document the FNO functions that can be seen in the FNO tab of your demo page, and name the parameters in a more sensible way. Eg from this description it's completely impossible to tell what this function expects, or what does it do:
        rdfs:label   "personInformationExtraction" ;
        fno:expects  ( <java:parameter.string.0> <java:parameter.string.1> <java:parameter.string.2> <java:parameter.string.3> ) ;
        fno:returns  ( <java:return.string> )
- "ss:javaScriptFilter": I don't think we need yet another syntax for functions or filtering.
  "record.get('(1,0).valueNumeric')[0] < 5" should be readable on its own, without the comment "iterate over papers that have less than 5 pages"
- Can your tool work with Open Office sheets? Google sheets? Maybe add to "Future work"
- About example 8: `String ifResult... ex:BestPaper` is not string but a constant term (in this case URL).
  And why `String elseResult ... ""` does not produce a triple like `?x rdf:type ""` ?
  I think that in this case <java:parameter.predicate.string.3> should be missing, rather than passing an empty string!

## About the Experimental Features
- ss:zip is a strange construct. It expects "predicateMap" to be a list of properties, and "objectMap" to return a list of values fitting those properties.
  The "ifRegexReturnGroup" function returns such list, produced from global regex matches over a string value (so it's badly named: there is no "if" in its work).
  It's not clear whether the more common case of splitting a cell to produce several values for the *same* property is covered by the tool.
- ss:Graph is also strange. It does not indicate how the data should be processed (that's hard-coded in fno.CustomFunctions.personInformationExtraction), nor what should be the resulting RDF (eg where it is specified that new UUID nodes should be made). That function is ad-hoc. It demonstrates handling multiple values (splits the cell on newline), but not how this can be done in general. I hope that conversion "domain experts" won't ever have to resort to writing functions in Java, so it'd be better to
- ss:SelectedObjects is not demoed in the examples

## Recommended Editorial Changes
- the "semi-structured data" you describe is actually structured (semi-structured would be if you need to handle Word documents, recognize headers from Excel typography, handle merged cells, etc)
- Reword "one of the tools which is RML Mapper"
- "meta data rich cells": maybe "meta data and rich cells"?
- "Besides it content" -> "Besides its content"
- "additional information like" -> "additional information such as"
- "Although, spreadsheets" -> remove comma
- "well-known RML-supporting tools like" -> "well-known RML convertors such as"
- "let us easy embed our new code" -> "lets us easily integrate our new code"
- "This removes the otherwise additional effort for mapping experts" -> "This eliminates extraneous effort for mapping experts"
- "This could be done in the R2RML implementation report" -> "This could be included in the R2RML implementation report"

---

### Meta-Review · Program_Chairs · 2021-04-21

**Recommendation:** Accept
**Confidence:** 5

**Metareview:**

The paper presents an extension of the RMLMapper tool to cover Excel features. The three reviewers agreed that the contribution is relevant to the workshop and presents a solid work. Please take into account the comments provided to include them in the camera-ready paper, try to be clear if the work presents a demo or a short research paper.

The recommendation is to accept.

---

### Decision · Program_Chairs · 2021-04-23

Accept